# The Effect of Piperidine Nitroxides on the Properties of Metalloproteins in Human Red Blood Cells

**DOI:** 10.3390/molecules28166174

**Published:** 2023-08-21

**Authors:** Stella Bujak-Pietrek, Anna Pieniazek, Krzysztof Gwozdzinski, Lukasz Gwozdzinski

**Affiliations:** 1Department of Physical Hazards, Nofer Institute of Occupational Medicine, 91-348 Lodz, Poland; stella.bujak@imp.lodz.pl; 2Department of Oncobiology and Epigenetics, Faculty of Biology and Environmental Protection, University of Lodz, 90-236 Lodz, Poland; anna.pieniazek@biol.uni.lodz.pl (A.P.); krzysztof.gwozdzinski@biol.uni.lodz.pl (K.G.); 3Department of Pharmacology and Toxicology, Medical University of Lodz, 90-752 Lodz, Poland

**Keywords:** nitroxide, red blood cells, oxidative stress, lactate dehydrogenase, methemoglobin

## Abstract

Nitroxides are stable, low molecular-weight radicals containing a nitroxide group that has an unpaired electron. The presence of a nitroxide group determines their redox properties. The effect of the piperidine nitroxides, Tempo, Tempol, and Tempamine, on metalloproteins (hemoglobin, superoxide dismutase, catalase) and lactate dehydrogenase in red blood cells was investigated in this research. In addition, the level of lipid peroxidation and the level of protein carbonyl groups were examined as indicators of the effect of oxidative stress. Nitroxides increased superoxide dismutase activity and oxidized hemoglobin to methemoglobin, and also slightly decreased the catalase activity of red blood cells treated with nitroxides. Tempol significantly decreased lactate dehydrogenase activity. All three nitroxides had no effect on membrane lipid peroxidation and protein oxidation. Our results confirm that nitroxides have both antioxidant and prooxidative effects in human red blood cells. The piperidine nitroxides do not initiate the oxidation of proteins and lipids in the membranes of human red blood cells.

## 1. Introduction

Nitroxides, also called nitroxyl radicals, are stable, low molecular weight, organic compounds containing a nitroxide group with an unpaired electron. The presence of a nitroxide group determines their antioxidant properties. However, under certain conditions, they can have pro-oxidant properties much like typical antioxidants. Nitroxides pass easily through the cell membrane and are non-immunogenic, non-toxic, and stable. Nitroxides exhibit significant anti-oxidant effects, including the ability to inactivate superoxide and peroxide, inhibit the Fenton reaction, and radical–radical recombination. It was found that nitroxides can reduce oxidative stress, affect the redox state of cells and tissues, and affect metabolic processes. These properties can be used for therapeutic and research purposes [1]. Recently, it has been shown that Tempo reacts with peroxide radicals, at a rate controlled by diffusion, in the presence of acids. Tempo can be regenerated from the oxoammonium ion by reaction with alkyl radicals. This reaction, a key step in synthetic transformations catalyzed by Tempo, takes place with k∼1–3 × 10^10^ M^−1^ s^−1^, thus allowing O_2_ to compete for alkyl radicals [2].

The antioxidant properties of nitroxides have been applied in biological and biomedical sciences. Many studies have shown that nitroxides prevent lipid peroxidation [3], protect cell DNA against oxidative damage [4,5,6], and show a protective effect in radiotherapy [7,8,9] and chemotherapy [7,9,10,11] in cancer. Nitroxides have been shown to significantly reduce reperfusion injury after temporary ischemia [12,13]. Nitroxides protected cardiomyocytes and B14 cells from the cytotoxic effects of doxorubicin [14,15,16]. They are also a part of the compound that initiates the apoptosis process in prostate tumor cells [17] or in Yoshida Sarcoma tumor cells [18].

Although nitroxides do not react with glutathione in vitro, in the presence of peroxynitrite, glutathione oxidation was enhanced and the EPR signal was lost. This result proves the pro-oxidative effect of nitroxides [19]. The pro-oxidative effect of Tempol was also demonstrated in the O_2_^•−^ inactivation induced by papain. As mentioned, nitroxide is oxidized to the oxoammonium cation in the presence of O_2_^•−^. This cation, due to its high reactivity, oxidized the -SH group of cysteine present in the catalytic center of papain [20].

Due to their functions, red blood cells (RBCs) are permanently exposed to reactive oxygen species (ROS) from both internal and external sources, and their oxygen metabolism is accompanied by the formation of superoxide and hydrogen peroxide (H_2_O_2_), and other ROS. The main endogenous oxidative source is the auto-oxidation of hemoglobin (Hb), which is bounded by a membrane and is hardly accessible to the cytosolic antioxidant system. This process deepens in hypoxic conditions. Partially oxygenated Hb shows an increased rate of autoxidation and even greater affinity to the red blood cell membrane [21]. Under regular conditions, Hb autooxidation in human RBCs is a physiological process, and approximately 3% of total Hb is converted to MetHb daily by slow oxidization by molecular oxygen [22].

Oxidative stress has been shown to play a significant role in damage to the structure and function of RBCs. It especially concerns the RBCs plasma membrane and impairs their deformability properties, which is crucial in microcirculation, when they have to deform while flowing through narrow capillaries. For this reason, RBCs are equipped with enzymatic and non-enzymatic antioxidant systems. Most ROS are inactivated by this antioxidant system, which includes superoxide dismutase (SOD), catalase (CAT), glutathione peroxidase (GPX), and peroxyredoxin-2 (PRDX2), and low molecular weight antioxidants such as glutathione, ascorbic acid, tocopherols, and others. The RBCs contain zinc–copper dismutase (Cu/ZnSOD, SOD-1, SOD), which scavenges the superoxide anion radical. There is a copper ion in the catalytic center of this enzyme, while zinc plays a structural and stabilizing role [23]. The reaction products of zinc–copper dismutase are oxygen and hydrogen peroxide, and dismutation rate constant is k = 6.4 × 10^9^ M^−1^ s^−1^. The SOD-1 and SOD-3 can be inhibited by cyanide or azide ions.

Another important RBCs enzyme is CAT, containing the heme in its catalytic center. CAT is an enzyme that breaks down hydrogen peroxide into water and molecular oxygen (k = 2 × 10^7^ M^−1^ s^−1^) [24]. The presence of CAT in the RBCs protects them against the toxic effects of hydrogen peroxide. At low H_2_O_2_ concentration, CAT may exhibit peroxidase activity and it can catalyze the oxidation of a wide range of organic and inorganic substrates, such as ethanol, methanol, nitrites, quinones, and others, by hydrogen peroxide. CAT inhibitors are hydrogen cyanide (HCN), hydrogen sulfide (H_2_S), and sodium azide (NaN_3_).

In turn, the lactate dehydrogenase (LDH) is an enzyme widely distributed in the body cells, belonging to intracellular oxidoreductases. LDH catalyzes the reaction between pyruvate and NADH, and the products of this reaction are lactate and NAD^+^. Cells have not been found to be able to actively clear the LDH. Therefore, it was considered that an increase in its activity and/or the activity of its individual serum isoenzymes in body fluids, such as cerebrospinal fluid or urine, is a result of damage to various tissues, in particular acute damage to the myocardium (e.g., infarction), RBCs, kidneys, skeletal muscles, liver, and lungs [25,26,27,28].

Many studies have shown that nitroxides can be used to protect cells and tissues against reactive oxygen species (in vivo and in vitro). However, the mechanism of their interaction with intracellular enzymatic and non-enzymatic antioxidant systems is not fully understood.

The aim of the study was to investigate the effect of the piperidine nitroxides Tempo, Tempol, and Tempamine (Figure 1) on metalloproteins (Hb, CuZnSOD, CAT) and LDH, which catalyzes the oxidation of pyruvate to lactate. We chose the metalloproteins that participate in redox cycles and the LDH oxidoreductase, and we determined the oxidative stress index using thiobarbituric acid.

Studies have shown that nitroxides affect metalloproteins, with the exception of CAT. Interestingly, nitroxides reduced LDH activity, but the results were statistically significant only for Tempol. All three nitroxides had no effect on the peroxidation of the membrane lipids and proteins.

## 2. Results

In this paper, we present studies on the influence of the three nitroxide radicals of piperidine derivatives on the chosen metalloproteins and the lactate dehydrogenase activity in the RBCs. In addition, due to the prooxidant and antioxidant properties of these compounds in RBCs, the basic index of oxidative stress was also examined. RBCs were selected for this study, because they are the most numerous cells among the morphotic blood components. Moreover, they are characterized by a relatively simple structure compared to other cells and have efficient antioxidant systems.

In our research, we used a 1 h-long sample incubation with the nitroxides. It has been previously shown that nitroxides can be found in RBCs after 30 s [29]. On the other hand, the half-time reduction of neutral piperidine nitroxides was reached after approximately 1 h [30].

Given that the nitroxides as free radicals can scavenge low molecular weight antioxidants, we first examined the ascorbic acid levels in RBCs treated with Tempo, Tempol, or Tempamine. The results are expressed as percent change from control (100%). A significant decrease in the level of the ascorbic acid was observed in RBCs treated with nitroxides (Figure 2). In the cells exposed to the nitroxide (0.2 mM), the level of this antioxidant decreased by 20–40% compared to the control cells. The increase in the nitroxide concentration resulted in a progressive loss of ascorbic acid in the cells. Thus, at a nitroxide concentration of 2 mM, only 30–40% of the initial pool of this antioxidant remained in the RBCs. These results confirm the fact that ascorbate plays an important role in the reduction of nitroxides.

The conducted research showed that 1-h incubation of RBCs with the selected nitroxides in doses above 0.2 mM causes a significant increase in the SOD activity (Figure 3). In addition, it was observed that the increase in the activity of this enzyme is proportional to the concentration of nitroxide. Depending on the nitroxide used and the concentration, the increase in the dismutase activity ranged from 55% to over 70%. However, 0.2 mM of Tempo did not significantly change the SOD activity in RBCs.

The second very important antioxidant enzyme of RBCs is CAT, which is responsible for the breakdown of hydrogen peroxide. The conducted studies showed that the activity of this enzyme in the nitroxides-treated RBCs did not change compared to the activity obtained for the control samples (Figure 4). Even the highest concentrations of the tested nitroxides did not cause significant differences in CAT activity compared to controls.

The LDH activity was determined in the hemolysate obtained from the RBCs after their incubation with the nitroxides. The obtained results showed that all three nitroxides slightly reduced the activity of the lactate dehydrogenase in the RBCs. However, a statistically significant decrease in LDH activity was observed only for the three highest concentrations of Tempol (Figure 5).

Another parameter determined in our study was the level of MetHb. The percentage of MetHb in the total Hb pool was determined spectrophotometrically. The conducted research showed an increase in the content of this metalloprotein in the RBCs treated with nitroxides (Figure 6). The greatest effect on the MetHb increase was shown by nitroxide Tempo (statistically significant results for the three highest concentrations). Tempamine had the least influence on the level of MetHb; only for the highest concentration of this nitroxide was a statistically significant increase in the level of MetHb observed.

To assess the degree of lipid peroxidation in RBCs incubated with nitroxides, the spectrophotometric method with thiobarbituric acid was used. Figure 7 shows the effect of the piperidine nitroxides on the degree of lipid peroxidation. The results are presented as percent change from the control samples. It was shown that none of the piperidine nitroxides used caused oxidative changes in the structure of lipids.

The next stage of our research included the evaluation of the degree of protein oxidation in RBCs treated with piperidine nitroxides. The level of protein carbonyl groups was determined using DNPH. Our studies showed that the value of the concentration of carbonyl groups of the RBCs proteins did not significantly change after their incubation with the tested piperidine nitroxides (Figure 8).

## 3. Discussion

Diffusing through biological membranes, the nitroxides are reduced to the corresponding hydroxylamines. Thiols play an important role in this process of nitroxide reduction in the cells [30]. The use of thiol group blockers (Hg(II), p-chloromercuric benzoic acid, and N-ethylmaleimide) led to a decrease in the rate of nitroxide reduction in the erythrocytes [30]. In one of the studies, we showed an influence of nitroxides on the glutathione level in human RBCs. The piperidine nitroxides decreased the reduced glutathione (GSH) level by up to 90% [31]. On the other hand, in the model systems, the GSH did not reduce nitroxides [32]. However, a decrease in the intracellular glutathione level in human RBCs was associated with the reduction of nitroxide to hydroxylamine. It was shown that ferricyanide inhibited the decrease in the GSH levels and the reduction of Tempo [33]. In our work, we also showed that the piperidine nitroxides led to a significant reduction in the activity of all glutathione-dependent enzymes tested, i.e., glutathione peroxidase, glutathione reductase, and glutathione S-transferase. The observed decrease in the enzymes activity was in each case dependent on the concentration of the nitroxide [34].

In addition to glutathione, ascorbic acid is another important water-soluble, low molecular weight antioxidant found inside RBCs. In reaction with the nitroxide, ascorbic acid is oxidized to the ascorbyl radical, which is rapidly disproportionated to dehydroascorbate. For this reason, nitroxides cause a significant reduction in the level of ascorbic acid in RBCs. In our study, we showed that the decrease in the concentration of ascorbate in RBCs depended on the concentration of the nitroxide. Beta-nicotinamide adenine dinucleotide and beta-nicotinamide adenine dinucleotide phosphate have been shown to increase the reducing potential of ascorbic acid in the reduction of nitroxides [35].

In our studies, we showed a significant decrease in the content of the ascorbic acid in erythrocytes after the nitroxide treatment. Tempo and Tempamine at 1 mM reduced the ascorbic acid levels below 20% of the control value, Tempol—below 10%, and at the highest concentration (2 mM) all three nitroxides—below 10% of the control value.

Nitroxide, similarly to SOD, catalyzes the dismutation of O_2_^•−^ to H_2_O_2_ and oxygen and is not consumed. Nitroxides exhibit pseudo-dismutase properties by dismutating the superoxide anion radical (O_2_^•−^) in a manner similar to SOD. In the case of SOD, the reaction proceeds in two steps. In the first stage, nitroxide is oxidized by O_2_^•−^ to oxoammonium cation, and in the second it is rapidly reduced by another O_2_^•−^ molecule to nitroxide [36,37]. The catalytic rate constants of the SOD mimetic activity increased at lower pH values. In addition, the rate constants were influenced by the redox midpoint potentials of the oxoammonium cation/nitroxide pair. The lower the midpoint potentials, the higher the rate constants, ranging from 10^6^ M^−1^s^−1^ to 10^8^ M^−1^s^−1^ [1]. Although the piperidine nitroxides have the highest pseudo-dismutase activity, the dismutation rate of O_2_^•−^ is approximately 1000 times slower than that of SOD [38].

By examining the effect of the nitroxides on the activity of SOD, we showed a significant increase in the activity of this enzyme depending on the concentration of the nitroxide used. This increase was significant for all three nitroxides. The observed increase in SOD activity was dependent on the concentration of the nitroxide and reached up to 70% of the control value. It was also shown in the model system whether nitroxide affects the activity of the commercial SOD preparation. The results of this experiment (not presented in the paper) confirmed the fact that nitroxides significantly increase dismutase activity. An increase in SOD activity was also observed in the erythrocytes of rats after the administration of vitamin E [39] or in animals exposed to the insecticide Propoxur (2-isopropoxyphenyl-N-methyl carbonate) [40]. The increased activity of SOD in vitro was also observed in human erythrocytes and lymphocytes under the influence of the biflavonoid silymarin [41].

It was shown that the irradiation of human skin fibroblasts with UVA1 radiation led to a decrease in their survival, a decrease in SOD activity, an increase in TBARS, and a decrease in collagen levels. The introduction of Tempol nitroxide before irradiation limited these changes and led to an increase in fibroblast survival [42]. Unlike nitroxide, Tempol-derived hydroxylamine was not protective in vitro, but in vivo it was protective against radiation-induced damage, which may be due to in vivo oxidation of hydroxylamine to nitroxide [43].

Recently, poly(Tempo) copolymer was shown to be superior to the free drug in scavenging O_2_^•−^ and reducing TNFα levels in an air sac model of acute local inflammation. In addition, the copolymer was reported to attenuate ROS levels after systemic injection in a footpad inflammation model. These data demonstrated the benefits of Tempo for in vivo efficacy in the treatment of inflammatory conditions [44].

In turn, lipophilic nitroxides and their reduced forms, non-paramagnetic hydroxylamines, can react with oxygen-centered and carbon-centered radicals to break the radical chain reaction. It was shown that lipophilic nitroxides were more effective in protecting membranes containing polyunsaturated fatty acids than α-tocopherol [3,45]. Water-soluble Tempo and Tempol showed a similar mechanism of action, protecting lipids against peroxidation initiated by ionizing radiation. Hydroxylamines have also been shown to be effective antioxidants [46]; unlike natural antioxidants, such as carotenoids and α-tocopherol, they do not show prooxidative activity [47].

Recently, it was shown that hydroxylamine derivatives of piperidine nitroxides effectively inhibited the oxidation and nitration of tyrosine, initiated by HRP/H_2_O_2_ and HRP/H_2_O_2_/nitrite. Interestingly, the piperidine hydroxylamines turned out to be such effective antioxidants as the corresponding nitroxides [48]. The combination of zeaxanthin with lipophilic nitroxides effectively protected against lipid peroxidation induced by singlet oxygen. The synergistic effect was related to the protective role of nitroxide in the intact structure of zeaxanthin, which enabled zeaxanthin to inactivate singlet oxygen. The nitroxides protected lipids as effectively as their reduced forms—hydroxylamines. The use of nitroxides increased the protective potential of zeaxanthin against AMD (age-related macular degeneration), compared to α-tocopherol, which is a natural antioxidant and defender of macular xanthophyll in the retina [49].

Nitroxides also exhibit pseudoperoxidase properties, stimulating the CAT activity of the heme proteins (Hb and myoglobin). It has been shown that these stable radicals can interact with Hb and myoglobin (Mb), significantly enhancing their CAT properties in the decomposition of hydrogen peroxide [50]. The catalytic activity induced by nitroxides is completely inhibited when glutathione is added [51]. In our studies, we showed a tendency towards a decrease in CAT activity; however, these results were not statistically significant. It has been shown that nitroxides can participate in the detoxification of hypervalent heme proteins such as ferrylmyoglobin (MbFeIV) [50]. Nitroxides, passing between two oxidation states, i.e., nitroxide and oxoammonium cation, lead to an increase in the activity of MbFeIII CAT mimicking, facilitating the decomposition of hydrogen peroxide and protecting biomolecules from hypervalent heme proteins [50]. Other antioxidant properties of nitroxides include the ability to oxidize transition metals that catalyze the Fenton and Haber Weiss reaction [52]. It is possible that this form, which is present in compound I catalase (FeIV), is inactivated by nitroxide, which causes a decrease in the activity of this enzyme.

The main internal source of ROS in RBCs is the autooxidation of Hb. The autooxidation of Hb leads to the generation of superoxides, but in this reaction MetHb is also produced. On the other hand, the superoxide hydrogen peroxide, formed as a result of dismutation, can react with both Hb and MetHb to give the appropriate ferryl form and ferryl radical form, in which iron is in the 4th oxidation state. Both forms show high reactivity with a biological material. Hb is bounded by a membrane and is hardly accessible to the RBC’s cytosolic antioxidant systems. This process is intensified in the hypoxic conditions, as partially oxidized Hb shows an increased rate of autooxidation and even greater affinity to the red blood cell membrane [21]. In our studies, we found an increase in the concentration of MetHb in the erythrocytes after incubation with Tempo, Tempol, and Tempamine. Oxidation of Hb to MetHb was the highest in the case of Tempo, because the concentration of 0.2 mM caused a significant increase in the level of MetHb. However, in the case of Tempol, a significant increase in MetHb occurred in the concentration of 1 mM. Tempamine had the least influence on the level of MetHb; only in the highest concentration of this nitroxide was a statistically significant increase in the level of MetHb observed. Our finding is consistent with the results of Balcerczyk et al., where Tempo caused the oxidation of the Hb to MetHb [33]. Hb oxidation by Tempo was completely inhibited by ferricyanide ((K_3_Fe(CN)_6_). Interestingly, (K_3_Fe(CN)_6_ is the Hb oxidant and should increase rather than inhibit the Hb oxidation. There are also studies which show that Hb oxidation was also found in trout erythrocytes after treatment with quinoline nitroxides [53].

In our studies, we showed that the piperidine nitroxides had different effects on hemoglobin oxidation in RBCs. The greatest effect was shown by Tempo, already from a concentration of 0.2 mM, and the smallest by Tempamine, only at a 10-times higher concentration. The nitroxides used differ significantly in terms of their hydrophobicity/hydrophilicity. The most hydrophobic is Tempo, which in a two-phase system is divided into the aqueous and lipid phases [54]. Tempol and Tempamine, on the other hand, are hydrophilic and have a similar character. However, nitroxides differ significantly in the electronegativity of the substituents present in the 4-position of the piperidine ring. Tempo shows the lowest electronegativity and Tempamine the highest [32]. In turn, comparing the reduction rate of the three nitroxides in the red blood cells, Tempamine is the fastest. It is reduced approximately three times faster than Tempol [30]. It seems that the stronger properties of Tempo in the oxidation of hemoglobin may result from its hydrophobic nature, which allows its penetration into the hydrophobic pocket in which there is heme in the hemoglobin molecule [55]. It is also possible that the faster reduction of Tempamine to hydroxylamine results in a decrease in the nitroxide concentration in the cell, which makes it ineffective in hemoglobin oxidation. In addition, it has been shown that in the presence of hemoglobin, hydrogen peroxide oxidizes the nitroxide to the oxoammonium ion, which has strong oxidizing properties and can oxidize Hb to MetHb [56].

The observed increase in SOD activity may lead to the production of larger amounts of hydrogen peroxide, which participates in the inactivation of LDH [57]. On the other hand, we did not observe a significant decrease in CAT activity, which indicates that H_2_O_2_ is removed by this enzyme. However, for a complete inactivation of H_2_O_2_, glutathione peroxidase is also needed, and the activity of this enzyme is reduced by nitroxides [34]. In addition, Hb oxidation is favored by a decrease in the level of glutathione and ascorbic acid, which enables the conversion of MetHb to Hb [58].

Another enzyme whose activity we investigated was lactate dehydrogenase. The LDH activity was determined using its ability to catalyze the conversion of pyruvate to lactate with the participation of NADPH. Based on the obtained results, it can be concluded that the incubation of RBCs with the nitroxides led to a decrease in LDH activity in the RBCs. However, statistically significant changes in the enzyme activity were observed only for Tempol in concentrations of 0.5 mM. LDH is an enzyme with a fairly high degree of stability. No inhibition of the enzyme was observed under the influence of cyanides or EDTA. However, it was found that LDH activity can be blocked by some of the metal cations, such as Hg(II), Cu(II) Zn(II), and Ni(II) [59]. In addition, the dehydrogenase activity is inhibited by iodides. Zhang et al. found that nitroxides lead to the release of LDH from cardiomyocytes induced by xanthine oxidase [16].

It was shown that Tempo led to the inactivation of alcohol dehydrogenase induced by AAPH (2,2’-azobis(2-amidinopropene) hydrochloride), as well as the degradation of deoxyribose initiated by tert-butyl hydroperoxide and ammonium persulfate [60]. These results testify to the prooxidative effect of nitroxides not only in human RBCs, but also in other cells [19,20,60]. On the other hand, Tempol was shown to protect blood plasma and platelets from oxidative damage induced by peroxynitrite. Exposure of platelets and plasma to peroxynitrite led to an increase in lipid peroxidation and carbonyl levels. Tempol has been shown to significantly inhibit lipid peroxidation and carbonyl formation in plasma and platelet proteins. In these studies, Tempol has been shown to have protective properties against oxidative damage induced by peroxynitrite [61].

We also investigated whether the applied nitroxides can initiate oxidative stress in RBCs. For this purpose, the level of substances reacting with thiobarbituric acid and the concentration of carbonyl groups were determined upon nitroxide treatment. We showed that the nitroxides do not initiate oxidative stress, causing lipid peroxidation in human red blood cells, and the Tempamine even showed a tendency to lower lipid peroxidation products.

## 4. Material and Methods

### 4.1. Chemicals

The following chemicals were purchased from Sigma Chemical Co. (St. Louis, MO, USA): Tempo (2,2,6,6-tetramethyl-1-piperidinyloxy), Tempol (4-hydroxy-2,2,6,6-tetramethyl piperidinyloxy), Tempamine (4-amino-2,2,6,6-tetramethyl piperidinyloxy). All other reagents of analytical purity were obtained from POCH S.A. (Gliwice, Poland).

### 4.2. Experiment Protocol

The experiments were conducted on RBCs isolated from the human blood buffy coat obtained from the blood bank in Lodz, Poland. The RBCs were washed three times with PBS (10 mM phosphate buffered saline, pH 7.4). The cells were suspended in Ringer’s solution to a hematocrit (Ht) of 10% and incubated for 1 h at room temperature with Tempo, Tempol, or Tempamine at a final concentration of 0.2 mM, 0.5 mM, 1.0 mM, or 2.0 mM. A control sample was included in each replication, consisting of RBCs incubated without nitroxide. After incubation, the samples were centrifuged and the erythrocytes were washed with 10 times the volume of cold PBS.

For the determination of enzyme activity, 1% hemolysate was prepared by adding 900 µL chilled distilled water to 100 µL washed erythrocytes (Ht 10%). The hemoglobin (Hb) concentration was estimated as cyanmethemoglobin using Drabkin’s reagent and absorbance measurements at 546 nm [62]. The enzyme activity measurements were performed on a Varian Cary UV-Vis spectrophotometer (USA) and other measurements on a LKB UV-VIS spectrophotometer (Sweden). In the manuscript, n-numbers represent samples from different donors.

### 4.3. Determination of Ascorbic Acid Level

To determine the concentration of the ascorbate in RBCs, a modified method described by Omaye et al. was used [63]. The principle of this method depends on the reduction of Fe^3+^ to Fe^2+^ by ascorbic acid. The Fe^2+^ and bathophenanthroline form a colored complex, whose level can be measured spectrophotometrically at λ = 525 nm. The concentration of the ascorbate was determined on the basis of a standard curve prepared for various concentrations of the ascorbate. The level of the ascorbic acid was expressed as a percentage of the control.

### 4.4. Superoxide Dismutase (SOD) Activity Assay

The activity of the SOD in RBCs was determined by the adrenaline method described by Misra and Fridovich [23]. This method is based on the ability of the SOD to inhibit the autooxidation reaction of adrenaline to adrenochrome occurring in an alkaline environment. The concentration of formed colored adrenochrome was measured at λ = 480 nm. The SOD activity was expressed as U/(mg Hb/min).

### 4.5. Determination of Catalase (CAT) Activity

The catalase activity was determined according to the method described by Aebi [64]. This method measures the rate of decomposition of hydrogen peroxide by CAT. The decomposition rate of the hydrogen peroxide was determined at λ = 240 nm. A decrease in the hydrogen peroxide absorbance by 0.036 units/min is the amount of CAT that breaks down 1 μmol of H_2_O_2_, corresponding to one unit of activity. The CAT activity was expressed in U/(mg Hb/min).

### 4.6. Determination of Lactate Dehydrogenase (LDH) Activity

The LDH activity in RBCs was determined by the method described by Wroblewski and Ladeu [65]. The LDH catalyzes the conversion of pyruvate to lactate, where reduced NADH is the proton donor. The applied method is based on the determination of the oxidation rate of NADH to NAD^+^. The rate of NADH oxidation was determined at the wavelength λ = 340 nm. The LDH activity was expressed in U/(mg Hb/min).

### 4.7. Determination of Thiobarbituric Acid Reactive Substances (TBARS) Level

The degree of lipid peroxidation in RBCs was determined using thiobarbituric acid (TBA), as described by Stocks and Dormandy [66]. The method is based on the reaction of thiobarbituric acid with lipid peroxidation products at low pH. The product of this reaction is optically active with maximum absorption at λ = 532 nm. The level of TBARS was expressed as a percentage of the control. The concentration of the TBARS was calculated using the millimolar absorption coefficient (156 mmol^−1^·cm^−1^) and expressed in nmol/mg Hb.

### 4.8. Evaluation of Carbonyl Group Concentration

The level of the carbonyl groups in the proteins of the RBCs was determined using 2,4-dinitrophenylhydrazine (DNPH) by the method described by Levine et al. [67]. The principle of this method is based on the reaction of DNPH with carbonyl groups, during which optically active (λ = 370) dinitrophenylhydrazones (DNP) are formed. The concentration of the carbonyl group was calculated using the millimolar absorption coefficient (22 mmol^−1^·cm^−1^) and expressed in nmol/mg Hb.

### 4.9. Determination of the Degree of Hb Autooxidation (% MetHb)

The percentage of MetHb in the total Hb pool was determined spectrophotometrically. The absorbance of the hemolysate was measured at two wavelengths: λ = 630 and λ = 700 nm. Then the potassium ferricyanide was added to the samples, converting all the Hb to MetHb. The absorbance of the samples was measured again at the same wavelengths. The percentage of MetHb was calculated using the formula below:$$\%\;MetHb=\frac{A_{630}-A_{700}}{A_{630^{*}}-A_{700^{*}}}\times 100\%$$
where: *A*—the absorbance of pure hemolysate, *A*_*_—the absorbance of hemolysate with potassium ferricyanide

### 4.10. Statistical Analysis

All data were expressed as mean ± standard deviation, calculated from the results of eight independent replicates. Normality of data was tested using the Shapiro–Wilk test, and the homogeneity of variance was verified with Levene’s test. The one-way repeated measures ANOVA and Tukey’s Post Hoc Multiple Comparison Test were used to estimate significance differences between groups. Statistical significance was accepted at *p* < 0.05 at the least. In all statistically significant comparisons, the power of the test was above 80%. The statistical analysis was performed using Statistica v. 13.3 (StatSoft Polska, Krakow, Poland).

## 5. Conclusions

Our results confirm that nitroxides have both antioxidant and prooxidative effects in human RBCs. Nitroxides reduce the level of low molecular weight antioxidants such as glutathione (which was found in earlier studies) and ascorbic acid. On the one hand, they cause a significant increase in SOD activity, and on the other hand, they reduce LDH activity and lead to an increase in the MetHb content. Piperidine nitroxides do not initiate the oxidation of proteins and lipids in the membranes of human RBCs.

## Figures and Tables

**Figure 1 molecules-28-06174-f001:**
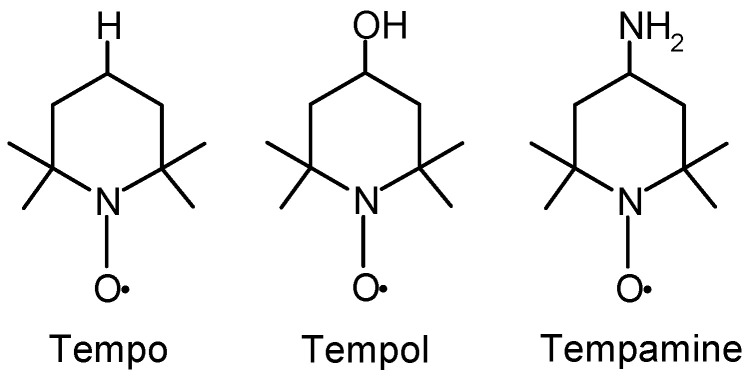
The chemical structure of Tempo, Tempol, and Tempamine.

**Figure 2 molecules-28-06174-f002:**
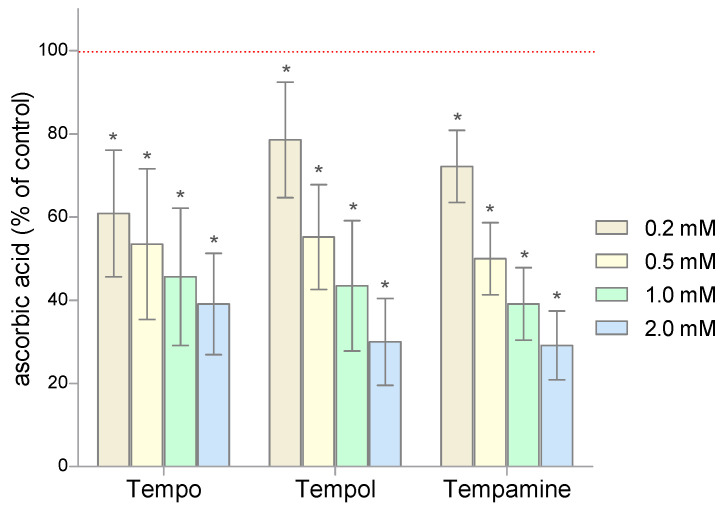
Changes of the ascorbic acid level in human RBCs after 1 h incubation with increasing concentration of nitroxides (n = 8). (* significantly different in comparison to control at least at *p* < 0.05).

**Figure 3 molecules-28-06174-f003:**
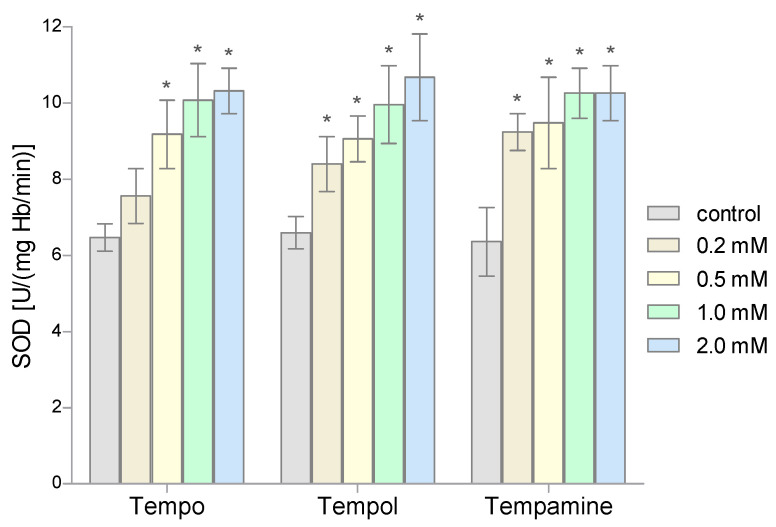
Changes of the superoxide dismutase activity in human red blood cells after 1 h incubation with the increasing concentration of nitroxides (n = 8). (* significantly different in comparison to control at least at *p* < 0.05).

**Figure 4 molecules-28-06174-f004:**
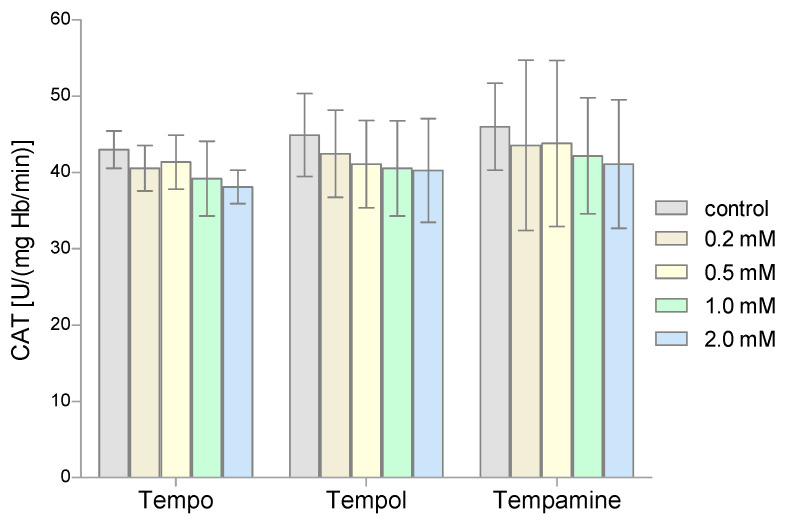
Changes of catalase activity in human red blood cells after 1 h incubation with the increasing concentration of nitroxides (n = 8).

**Figure 5 molecules-28-06174-f005:**
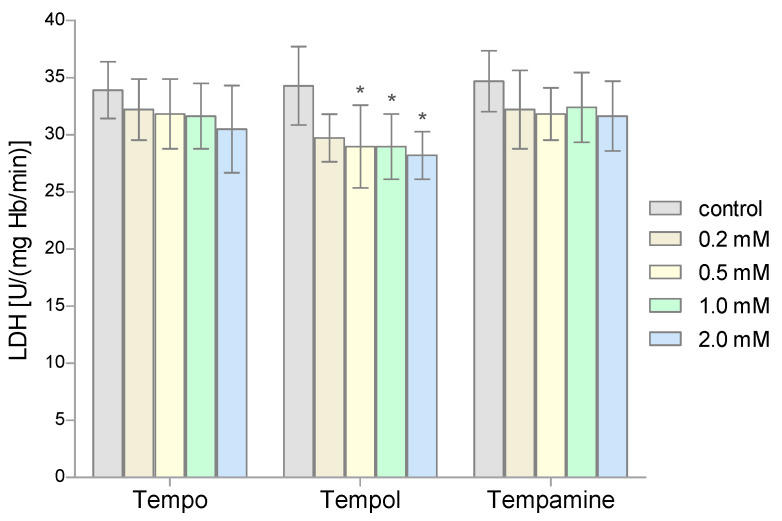
Changes of lactate dehydrogenase activity in human red blood cells after 1 h incubation with the increasing concentration of nitroxides (n = 8). (* significantly different in comparison to control at least at *p* < 0.05).

**Figure 6 molecules-28-06174-f006:**
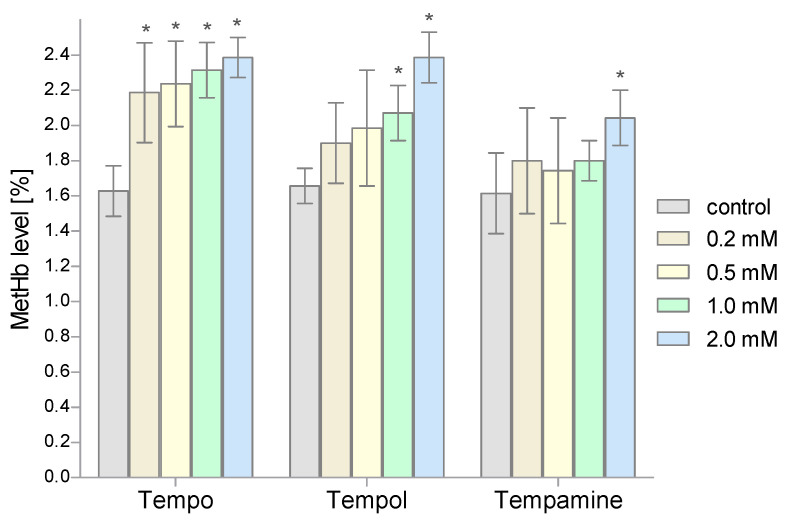
Changes of the methemoglobin level in human red blood cells after 1 h incubation with increasing concentration of nitroxides (n = 8). (* significantly different in comparison to control at least at *p* < 0.05).

**Figure 7 molecules-28-06174-f007:**
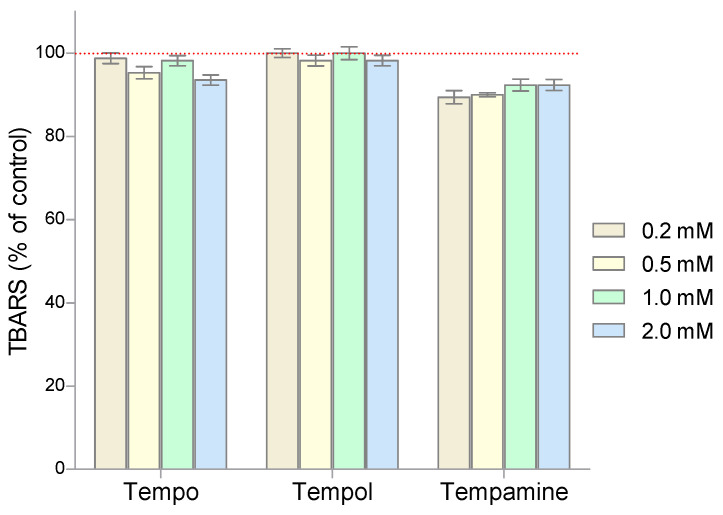
Changes of TBARS level in human red blood cells after 1 h incubation with increasing concentration of nitroxides (n = 8).

**Figure 8 molecules-28-06174-f008:**
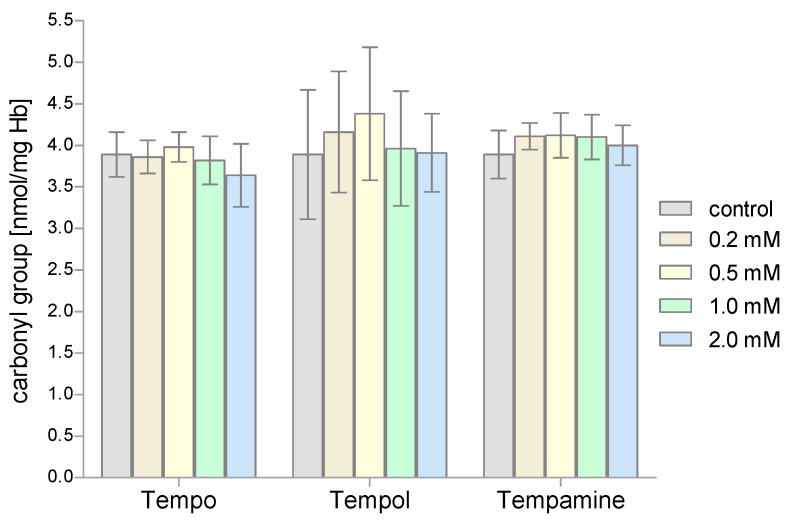
Changes of the carbonyl group concentration in human red blood cells after 1 h incubation with increasing concentration of nitroxides (n = 8).

## Data Availability

Not applicable.

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
