# Peer review of "The Effect of Piperidine Nitroxides on the Properties of Metalloproteins in Human Red Blood Cells"

_molecules, 2023, doi:10.3390/molecules28166174_

Round 1

Reviewer 1 Report

In the manuscript: “The effect of piperidine nitroxides on the properties of metallo-proteins in the human red blood cells” by Stella Bujak-Pietrek et al. the authors describe the effect of some piperidine nitroxides on metalloproteins and LDH from red blood cells. Moreover, they analyzed lipid peroxidation, protein carbonylation, and meta-hemoglobin level as indicators of oxidative stress concluding that piperidine nitroxides have both prooxidant and antioxidant activity.

In a previous paper, the authors showed that nitroxides decreased GSH levels, in this manuscript they showed that nitroxide treatment decreased also ascorbic acid levels. On the other hand, the treatment increases the SOD activity and does not have an effect on CAT activity, or on LDH activity while increasing the levels of meta-hemoglobin. Finally, piperidine nitroxide treatment has no effect on lipid peroxidation or on protein carbonylation.

The topic of this manuscript is very interesting, several studies describe the antioxidant properties of nitroxides that are mainly attributed to their ability to undergo reversible redox reactions, nevertheless, their mechanism of action is still unclear.

The results presented are coherent with the conclusion presented, I have only a question :

The authors state that the oxidation of Hemoglobin to meta-hemoglobin is due to hydrogen peroxide generated by the dismutation of oxygen superoxide. They found that the three nitroxides used have a different impact on hemoglobin oxidation with the lowest effect exerted by tempamine. Since the effect of nitroxides on SOD is the same, can the author give some explanation for this difference?

Reviewer 2 Report

The relationship of nitroxides with those of the membrane lipid peroxidation and protein oxidation is complicated, especially in cells. It is significance to explore the effect of nitroxides on both antioxidant and prooxidative functions in the human red blood cells. In this work, the effect of the piperidine nitroxides: Tempo, Tempol, and Tempamine on metalloproteins (hemoglobin, superoxide dismutase, catalase) and lactate dehydrogenase in red blood cells was studied. Moreover, the level of lipid peroxidation and the level of protein carbonyl groups were analyzed as indicators of the effect of oxidative stress, the studies is helpful for understanding the effect of nitroxides on the oxidation of proteins and lipids in the membranes of the human red blood cells. However, some revisions should be considered for the manuscript.

1. For the determination of enzyme activity (525 nm, 480 nm, 240 nm), reactive substances level (532 nm), carbonyl group concentration (370 nm), and degree of Hb autooxidation (630 nm, 700 nm), the measurements were performed at different wavelength. The model of  spectrometer should be showed in Experiment protocol. And it is better the change of the the absorption spectrum were listed in Supporting Information as Supplementary material.

2. The molecular structure of Tempo, Tempol, and Tempamine could be shown in the Introduction as Scheme 1.

3. The language could be refined, and Figure 7is Figure 1 in Page 4. 

The language could be refined, and Figure 7” is Figure 1 in Page 4.
